# Concepts and Key Technologies of Microelectromechanical Systems Resonators

**DOI:** 10.3390/mi13122195

**Published:** 2022-12-11

**Authors:** Tianren Feng, Quan Yuan, Duli Yu, Bo Wu, Hui Wang

**Affiliations:** 1College of Information Science and Technology, Beijing University of Chemical Technology, Beijing 100029, China; 2Guangdong Institute of Semiconductor Micro-Nano Manufacturing Technology, Foshan 528000, China

**Keywords:** MEMS, resonator, oscillator

## Abstract

In this paper, the basic concepts of the equivalent model, vibration modes, and conduction mechanisms of MEMS resonators are described. By reviewing the existing representative results, the performance parameters and key technologies, such as quality factor, frequency accuracy, and temperature stability of MEMS resonators, are summarized. Finally, the development status, existing challenges and future trend of MEMS resonators are summarized. As a typical research field of vibration engineering, MEMS resonators have shown great potential to replace quartz resonators in timing, frequency, and resonant sensor applications. However, because of the limitations of practical applications, there are still many aspects of the MEMS resonators that could be improved. This paper aims to provide scientific and technical support for the improvement of MEMS resonators in timing, frequency, and resonant sensor applications.

## 1. Introduction

With the rapid development of the electronics industry and Industrial Revolution 4.0, microelectromechanical systems (MEMS) technologies are playing an increasingly important role in applications, such as sensing [1], filtering [2], frequency reference [3], bio-diagnostics [4], and energy harvesting [5]. It is becoming a steadily growing multi-billion dollars industry [6]. The core components of MEMS are typically small, micron-sized moving mechanical parts that rely on energy conversion between mechanical and electronic domains to perform functions, such as sensing and energy harvesting. The frequency reference application relies on the vibration of the MEMS resonator to constitute the oscillator [7]. MEMS resonators have always been an important topic in the field of vibration engineering. By 2024, it is anticipated that the market for MEMS resonators would have increased by a factor of six, reaching $600 million [8]. Recently MEMS resonators have shown a similar performance to quartz crystal resonators. This has caused the MEMS resonator to become an attractive solution for performance-enhancement.

Quartz crystal resonators have dominated the market [9]. However, quartz crystals are obtained by conventional individual cutting techniques [7], and their size is difficult to reduce [10]. In addition, quartz crystals are not compatible with CMOS processes [11] and their integration [11], reliability [12], and power potential [13] are unsatisfactory. Recently, researchers have been working to replace conventional quartz resonators with silicon-based resonators because of their small size [3], high reliability [14], good compatibility with CMOS processes [15], and low-cost batch manufacturing [16]. MEMS resonators have been reported to have excellent long-term stability [10], high quality factors, and high reliability. Recently, some relevant reviews have been published: Ref. [17] discusses the vibration modes of MEMS resonators, simplified models, and their applications; Ref. [18] provides an overview of the fabrication methods for silicon-based MEMS resonant sensors; Ref. [19] presents a review of recent advances in resonator-based M/NEMS logic devices; Ref. [20] discusses the piezoelectric resonator materials, process flow, and performance improvement methods; and Ref. [21] mainly reviews the dissipation analysis methods and quality factor enhancement strategies of piezoelectric MEMS lateral vibration resonators. However, because of the limitations of practical applications, there are still many aspects of MEMS resonators that could be improved. The resonant frequency, quality factor, frequency accuracy, electromechanical coupling coefficient, motional resistance, and temperature stability are important performance indicators of MEMS resonators. This article focuses on reviewing the entire process of MEMS resonators in timing, frequency, and resonance sensing applications from preliminary design to optimization of important indicators and engineering corrections. It aims to provide some design references for improving the performance of MEMS resonators. 

This paper has the following format. Section 2 describes the operating principles of MEMS resonators. Section 3 explains the main technologies for MEMS resonators. The summary and future perspective are presented in Section 4.

## 2. MEMS Resonator Operation Principle

MEMS mechanical structures typically operate at surface, bending, torsional, and bulk modes in timing, frequency, and resonant sensor applications. The equivalent model, vibration modes, and transduction mechanisms are described in the following.

### 2.1. Equivalent Model

As shown in Figure 1a, a mass damper-spring system can be used to express the MEMS resonator as [7]:(1)$$m_{eq}\frac{\partial^{2}x}{\partial t^{2}}+c_{eq}\frac{\partial x}{\partial t}+k_{eq}x=F_{e}$$
where $m_{eq}$, $c_{eq}$, $k_{eq}$, x, $F_{e}$, are the equivalent mass, equivalent damping factor, equivalent stiffness, displacement, and external excitation in the system, respectively.

As shown in Figure 1b, a MEMS resonator is able to represented by the Butterworth Van-Dyke (BVD) model as:(2)$$L_{m}\frac{\partial i}{\partial t}+R_{m}i+\frac{1}{C_{m}}\int i\partial t=\upsilon$$
where $L_{m}$, $R_{m}$, $1/C_{m}$, i, υ are the equivalent inductance, equivalent resistance, equivalent capacitance, current, and voltage in the system, respectively. Sometimes the feedthrough capacitance $C_{0}$ is introduced to take account of parasitic effects.

As a typical second order system, the resonant frequency of the resonator can be calculated as:(3)$$f_{0}=\frac{1}{2\pi }\sqrt{\frac{k_{eq}}{m_{eq}}}=\frac{1}{2\pi }\sqrt{\frac{1}{L_{m}C_{m}}}$$

The electromechanical coupling factor η can be defined to relate the electrical component to the mechanical device. The mapping relationship is calculated as: $L_{m}=m_{eq}/\eta^{2}$, $R_{m}=c_{eq}/\eta^{2}$, $C_{m}=\eta^{2}/k_{eq}$.

### 2.2. Vibration Modes

Most resonators vibrate in flexural mode (in-plane, out-plane) [22], plate wave mode (lamb, shear-horizontal), torsional mode, and bulk mode (contour mode, thickness mode, shear mode). The diagrams of basic vibration modes are shown in Figure 2. Table 1 summarizes the frequency calculation formulas for common vibration modes.

As shown in Figure 2a–d, the flexural mode is characterized by low acoustic velocity and large vibration displacement, which usually appears in MEMS resonators with the cantilever beam or membrane structures. The flexural modes are generally classified as in-plane or out-of-plane modes for low-frequency applications.

MEMS resonators form a bulk mode by expanding or contracting. The bulk mode has a greater stiffness, acoustic velocity, and resonant frequency. The categories of bulk modes can be roughly divided into contour modes, thickness modes, and shear modes. Contour modes are commonly found in resonators with plate or disc structures and are mainly classified as length extensional mode (LE), width extensional mode (WE) [23] and radial breathing mode (radial breathing) [24]. As shown in Figure 2e–g, the resonant frequency depends on the transverse physical dimensions of the structure [25]. 

Since the contour mode resonators with different frequencies can be defined in a single wafer by lithography, contour mode resonators are often found in multi-frequency integration applications. 

The most typical MEMS device for the thickness mode is the thin film bulk acoustic resonator (FBAR), whose resonant frequency depends on the thickness of the resonator. FBARs have been widely used as filters in radio frequency (RF) systems due to their greater coupling and low motional resistance [26].

The total volume of the shear mode resonator is constant because of the simultaneous expansion and contraction. The shear mode is generally observed in square and circular resonators, and is mainly classified as lamé mode and face-shear mode (FS). 

Table 2 presents representative resonators with different modes in specific applications. 

### 2.3. Transduction Mechanisms

Capacitive and piezoelectric are the most common transduction mechanisms for MEMS resonators. The capacitive resonator can be viewed as a simple structure. The resonator is biased by a DC voltage and then excited by the upper and lower plates with a specific frequency signal. The vibration of the resonant structure causes a change in electrical capacity and generates an induced current. The piezoelectric MEMS resonator relies on the piezoelectric material and effect to perform the energy conversion.

Capacitive MEMS resonators are generally composed of single crystal or polycrystalline silicon. Relying on the extremely low material losses of silicon, capacitive MEMS resonators have a high quality factor. However, capacitive resonators have a low electromechanical coupling factor and high motional resistance, which can be calculated as:(4)$$\eta =\nabla C\cdot V_{p}$$
(5)$$R_{m}=\frac{\sqrt{k_{eq}m_{eq}}}{Q}\frac{g^{4}}{V_{p}^{2}\varepsilon^{2}A^{2}}$$
where η is the electromechanical coupling factor, ∇C is the capacitive gradient between the two electrodes, $V_{p}$ is the DC voltage applied to the MEMS resonator, $R_{m}$ is the motion resistance, Q is the quality factor, g is the capacitive MEMS resonator gap width, ε is the capacitive gap permittivity, and A is the effective transduction area.

Piezoelectric MEMS resonators, such as thin-film piezoelectric substrate resonators (TPoS) and FBARs, generally consist of electrodes, piezoelectric materials, and substrates. Relying on the piezoelectric effect of piezoelectric materials, piezoelectric MEMS resonators have high η and low $R_{m}$ without $V_{p}$. Taking the single-port WE mode MEMS piezoelectric resonator as an example, the η and $R_{m}$ can be calculated as:(6)$$\eta =2d_{31}E_{eq}L$$
(7)$$R_{m}=\frac{\pi }{8}\frac{T}{L}\frac{\sqrt{\rho_{eq}}}{QE_{eq}^{\frac{3}{2}}d_{31}^{2}}$$
where $d_{31}$ is the piezoelectric coefficient.

Table 2 presents representative resonators with different types, Q, and transmissions. Capacitive MEMS resonators are advantageous because of their high Q, while piezoelectric MEMS have the advantage of low $R_{m}$. 

In addition, thermal and magnetic excitations may be not as strong as the mentioned mechanisms, but they have received much attention. Magnetically excited MEMS resonators form stable oscillations through magnetoelectric coupling and magnetostrictive effects [32]. Compared with electrostatic and piezoelectric excitation, the magnetic field can be applied from a distance, so the structural design of magnetically actuated MEMS resonators is more variable [33]. In particular, magnetically actuated MEMS resonators can be used in antennas with impedance matching of 50 Ω; the volume of the device is several orders of magnitude smaller than that of traditional antennas. Some MEMS resonators are applied to low frequency applications through thermal driving [34]. Thermally driven resonators typically rely on the piezoresistive effect for signal detection [34]. Thermally driven resonators have the strong driving ability, small film damping, and simple structure [35]. By subtracting the reference signal from the sensor output, Ref. [36] has successfully suppressed an asymmetric resonance in a thermal piezoresistive cantilever sensor. The results show that the proposed sensor obtains a quality factor of approximately 1893 and exhibits a shorter response time.

In timing applications, phase noise has become a major factor limiting circuitry. The short-term stability is mostly determined by phase noise, and the single-sideband phase noise (Figure 3) in an oscillator quantified by Leeson’s equation is expressed as [7]:(8)$$L(\Delta f)=\left(\frac{2Fk_{b}T_{e}}{P_{0}}+\frac{b_{1}}{\Delta f}\right)\left(1+\frac{f_{c}^{2}}{4Q^{2}}\frac{1}{\Delta f^{2}}\right)$$
where Δf is the frequency offset, F is the noise figure, $k_{b}$ is the thermal noise figure, $T_{e}$ is the temperature, $P_{0}$ is the signal power, $b_{1}$ is the correction factor, and $f_{c}$ is the center frequency.

Refer to Equation (8), the capacitive MEMS oscillator rely on the high Q resonator to achieve low phase noise, whereas the piezoelectric MEMS oscillator allows it to handle greater signal power by its low $R_{m}$. Table 2 presents representative resonators with different transduction mechanisms in specific applications. 

## 3. Performance and Optimization

This section reviews the key performance indicators of MEMS resonators in timing, frequency, and resonant sensor applications, mainly including quality factor Q, motional resistance $R_{m}$, frequency consistency, and temperature stability. Then, the optimization methods indicators are explained below.

### 3.1. Quality Factor

Refer to Equation (8), the Q is a critical performance for MEMS resonators. There are various definitions of Q. The Q describes the ratio between the energy stored and the energy dissipated in a cycle, which can be calculated as:(9)$$Q=2\pi \frac{E_{store}}{E_{loss}}=\frac{i^{2}X_{L_{m}}}{i^{2}R_{m}}=\frac{X_{L_{m}}}{R_{m}}=2\pi f_{0}\frac{L_{m}}{R_{m}}=\frac{f_{0}}{\Delta f}$$
where i is the current of the RLC circuit, $X_{L_{m}}$ is the imaginary part of the impedance, and $R_{m}$ is the real part of the impedance.

The quality factor in engineering is defined as the ratio of the center frequency to the −3 db bandwidth. It can be seen that the lower the energy loss, the higher the quality factor. Usually, the main energy loss of a MEMS resonator consists of air damping loss, anchor loss, thermoelastic loss, and other losses. The quality factor is affected by all the above losses and can be expressed as:(10)$$Q={\left(\frac{1}{Q_{air}}+\frac{1}{Q_{anchor}}+\frac{1}{Q_{TED}}+\frac{1}{Q_{others}}\right)}^{-1}$$

#### 3.1.1. Air Damping Loss

The air damping loss [37] 
can be neglected in the macroscopic domain. However, since the 
surface-to-volume ratio of MEMS resonators becomes larger, the air damping 
become non-negligible. Although some estimation methods have been proposed for *Q_air_*, 
the predictions sometimes vary widely because air damping is affected by 
resonator size, gap size, ambient pressure, vibration modes, and non-ideal 
fluid motion. Several estimation methods are summarized below.

When based on the molecular regime method, the $Q_{air}$ of wide plates can be calculated as:(11)$$Q_{air}∼{\left(\frac{t}{L}\right)}^{2}/P_{air}$$

When based on incompressible unbounded fluid method, the $Q_{air}$ can be calculated as:(12)$$Q_{air}∼\frac{t^{2}W}{{\left(\mu L\right)}^{2}}$$

When based on incompressible squeeze film method, the $Q_{air}$ can be calculated as:(13)$$Q_{air}∼{\left(\frac{t}{wL}\right)}^{2}\frac{g^{3}}{\mu }$$
where $P_{air}$ is the ambient pressure, μ is the viscosity coefficient that is positively related to the ambient pressure, and g is the resonator gap width.

Table 2 compares the quality factors of the resonators at different ambient pressures. It can be known that $Q_{air}$ is related to the resonator structure, resonance mode, frequency, and ambient pressure. Air damping has a greater effect on lower frequency resonators than high frequency resonators, and when the ambient pressure exceeds a certain vacuum level, the air damping loss can be negligible. Therefore, making MEMS resonators work in a vacuum environment is the most common method. Furthermore, Ref. [38] proposes a method to perforate the lower electrode of a clamped-clamped microbeam resonator to reduce the effects of squeezed membrane damping and allow operation at atmospheric pressure.

#### 3.1.2. Anchor Loss

When the resonator is vibrating, elastic waves propagate to the substrate through the anchor point. This loss is called anchor loss or support loss. Studies have shown that anchor loss is affected by frequency, resonant mode, resonator size, anchor location, and elastic wave transmission conditions. It can be estimated as follows.

For the plane mode resonator, the $Q_{anchor}$ can be calculated as:(14)$$Q_{anchor}∼{\left(\frac{L}{W}\right)}^{3}$$

For the out of plane mode resonator, the $Q_{anchor}$ can be calculated as:(15)$$Q_{anchor}∼\frac{L}{W}{\left(\frac{h}{t}\right)}^{2}$$
where h is the substrate thickness.

For resonators with more complex structures, researchers often predict anchor loss in finite element analysis software. For double-clamped cantilever resonators, the anchor loss is usually considered to be larger as the resonant frequency increases. For low-frequency (<200 MHz) aluminum nitride (AlN) contour mode resonators at a low temperature (<25 °C), anchor loss is the main loss mechanism. For resonant modes with minimal displacement nodes, such as free–free beam and lateral mode resonators, using the minimum displacement nodes as an anchor can minimize the energy leakage.

The main methods to reduce anchor loss include the quarter-wave tether method, the acoustic reflection method, and the phononic crystal (PnC) method. The quarter-wave tether method refers to setting the transmission tether to be a quarter wavelength so that the acoustic wave can be reflected. Ref. [39] proposed a “hollow-disk” ring resonator with cross-support beams. The anchor point of the support beam center reflects the wave to the ring resonator and raises the quality factor to 10000. Acoustic reflection methods generally reflect elastic waves through etched trenches in the substrate. To test resonators with different depths and widths of trenches, Ref. [40] set trenches around disk-shaped resonators to reflect surface waves. The result shows that the proposed structure can increase the quality factor by four times. A quality factor of 4522 was achieved by [41] after adding a T-shaped tether with reflective blocks to the 10 MHz lateral mode resonator. A method to effectively reduce anchor loss by introducing a slot near the tether support end of a lamb-mode piezoelectric resonator was proposed by [42]. The results show that the designed 1.97 GHz resonator achieves a Q value of 3140. The PnC method is to embed PnC around the resonator and prevent acoustic leakage. By tuning the PnC array to form an extremely narrow-band filter, the outward propagation of elastic waves can be suppressed. By introducing a cross-shaped 2D PnC outside the anchor, Ref. [43] increased the mass factor from 21,180 to 221,536. To suppress the displacement to reduce energy loss, Ref. [29] proposed to use the PnC-reflector composite structure. The results show that the combination of PnC and reflector provides a quality factor of up to 4682. A disk-shaped 2D PnC matrix was deployed at the anchor point of the resonator by [44], thereby increasing the quality factor from 2572 to 9242. A butterfly-shaped structure resonator, a resonator with PnC on tethers and a resonator with PnC on anchors were proposed by [45]. The results show that the quality factors of the proposed three 170 MHz resonators are 51503, 83349 and 83899, while the traditional resonator quality factor is only 29899. A 133 MHz TPoS MEMS resonator with a PnC-strip-anchor-tie-line was proposed by [46], and the cell number on the quality factor was evaluated. The results show that the quality factor of the resonator with the proposed PnC strip is 19,902. A 52 MHz AlN-on-SOI MEMS resonator with a suspended frame structure and a PnC was proposed by [30]. The suspended frame structure isolates mechanical vibrations between the resonator and the substrate, while the PnC array acts as a frequency-selective reflector to reduce energy leakage. The results show that the quality factor of the proposed structure reaches 4743, which is 7.8 times higher than that of the traditional MEMS resonator. A similar structure was proposed by [47]. Simulation results show that the proposed method reduces the anchor loss of the 90 MHz ring piezoelectric resonator with a quality factor as high as 100,000, which is 83 times higher than the original structure. There are other PnC structures proposed, such as spider web-shaped PnC [48] and PnC with framing holes stub [49]. The PnC method is effective in reducing anchor loss but may increase the fabrication complexity. Table 3 shows some representative methods and diagrams for reducing anchor loss.

#### 3.1.3. Thermoelastic Loss

Due to the thermal expansion coefficient (CTE) of silicon, the temperature of the compression region increases and the temperature of the tensile region decreases when the resonator vibrates. The temperature difference generates heat flow, which leads to energy dissipation. It is called thermoelastic damping (TED) [53]. Zener developed a general expression for thermoelastic damping with flexural mode MEMS resonators as follows [54]:(16)$$Q_{TED}=\frac{\rho C_{p}}{E\alpha^{2}T}\frac{1+{\left(\omega \tau \right)}^{2}}{\omega \tau }$$
(17)$$\tau =\frac{\rho C_{p}}{\kappa }\frac{t}{\pi }$$
where $C_{p}$ is the specific heat at constant pressure, α is the thermal expansion coefficient, ω is the resonant angular frequency, τ is the thermal time constant, and κ is the thermal conductivity.

A. Duwel proposed the fully coupled thermodynamic equations and uncoupled thermodynamic kinetic equations to calculate the thermoelastic damping. It was demonstrated that the TED of the fundamental longitudinal mode and torsional mode resonator is negligible [55]. Chandorkar showed that since the deformation of the torsional mode is isovolumic and therefore not limited by TED [56]. A. Duwel demonstrated that the effect of TED can be reduced by moving the thermoelastic debye-resonance away from the operating frequency [57]. R. N. Candler proposed to reduce the TED effect of a 610 kHz flexural mode resonator by adding the slot near the anchor point [58]. As shown in Table 3, Ref. [50] affected thermo-mechanical coupling by adding slots to the clamped-clamped MEMS beam resonators, and tests the effects of different positions and sizes of the slot on the resonator. The result shows that a reasonable slot structure can improve the quality factor of the proposed resonator by 3000. A semi-analytical approach to predicting TED was introduced by J. Segovia-Fernandez. It was demonstrated that the quality factor of AlN contour mode resonator (CMR) resonators can be improved by optimizing the coverage area of the metal layer [59]. As pointed out by [27], for lamé mode plate resonators, etched holes will introduce an additional temperature gradient so that TED cannot be ignored. Therefore, a thermo-mechanical coupling equation was established to analyze the dependence of TED on the distribution of etched holes. A 2.81 × 10^19^ quality factor was obtained by optimizing the size and distribution of holes. That material, orientation, doping level, and slot location all affect the temperature stability of tuning fork MEMS resonators was experimentally demonstrated by [51]. Therefore, it is proposed that intelligent optimization algorithms, such as Covariance-Matrix-Adaptation-Evolution-Strategy can be used to determine the geometry of MEMS resonators to maximize quality factor and temperature stability. It is the belief of [60] that the traditional TED model is not suitable for partially coated resonators, so an analytical TED model of a partially covered cantilever with a silicon oxide coating was developed. The results show that the proposed TED model matches well with the finite element method. It is also pointed out that the length of the metal coating should be less than 70%, and the influence of TED will be reduced to 25% when the position of the metal coating is far from the clamping end.

#### 3.1.4. Other Losses

In addition to the main losses above, there are other losses including coating loss, electrical loss, and internal material loss. The coating loss is generally presents in resonators with metallic or piezoelectric layers. Friction between each layer causes energy loss as the resonator vibrates. H. Qiu concluded that coating loss may be the main reason that the quality factor of piezoelectric resonators is lower than that of capacitive resonators [52]. They fabricated piezoelectric and electrode coated cantilever beams with 20% to 100% length coverage. As shown in Table 3, the coating loss of a 20 kHz flexural mode resonator can be reduced by decreasing the coating coverage [52]. The result shows that the resonator with 20% coating coverage can increase the quality factor from 3000 to 8000 compared with the full coverage resonator. R Sandberg tested the quality factor of cantilevers with different gold-coated thicknesses and showed that the coating loss is proportional to the thickness of the coating. S. Dohn pointed out that the position of the coating also affects its quality factor. The experimental result shows that the quality factor can be effectively improved when the coating is placed on the tip of the cantilever [61]. The effect on the quality factor of AlN resonators when the top electrode is gold or aluminum was compared by [62]. Top electrodes with different thicknesses were also fabricated to analyze their effect on the resonator performance. The results show that when the top electrode material is gold and the electrode thickness is reduced from 1 μm to 0.5 μm, the quality factor increases from 9939 to 12983. In addition, Ref. [63] proposed to use of the lattice mismatch between GaN and Si to introduce greater stress in the GaN epitaxial layer to store elastic energy. The quality factor of the designed 911 kHz double-clamped resonator reaches 100,000. Ref. [64] believed that the stray modes of the resonator will inevitably damage the quality factor, and proposed to suppress the stray modes by optimizing the electrodes to distribute the charge density uniformly in the z-direction. The results show that the quality factor of the proposed resonator reaches 10,069 at 83.59 MHz, which is 2.7 times higher than the ordinary one. When the movement of charges is blocked by the ohmic loss, which was called the electrical loss. The electrical loss is generally independent of the design and fabrication of the resonator structure and can be ignored. The surface effect become significant due to the increased surface-to-volume ratio of the resonator [65]. Impurities, lattice defects, absorbers and other defects on the resonator surface can generate surface stress, resulting in surface loss or friction.

### 3.2. Motional Resistance

The motional resistance determines the power attenuation of the resonator. According to Barkhausen’s criteria [66], the loop gain required to sustain oscillation is determined by the motional resistance. The low motional resistance supports higher input signal power levels. Referring to Equation (8), increasing the input signal power can reduce the phase noise. In wireless communication applications, high motional resistance (>50 Ω) can hinder the deployment of resonators in radio frequency (RF) front-ends. 

According to Equations (6) and (7), resonators can obtain a low motional resistance and a high electromechanical coupling factor. This conclusion has been verified in studies, such as FBARs, shear mode quartz resonators, and contour mode AlN resonators. D. E. Serrano proposed that a piezoelectric resonator with a 22.5% electrode coverage could have the highest electromechanical conduction efficiency and low motional resistance [67]. The capacitive MEMS resonator rely on the high quality factor to achieve low phase noise, whereas the piezoelectric MEMS resonator can handle greater signal power by virtue of its low motional resistance. Ultimately the capacitive and piezoelectric MEMS resonator can achieve similar phase noise performance. Multiple resonators were combined into an array structure to improve the power processing capability by [68]. The result shows that the proposed array structure reduces the phase noise by 26 dB compared with the single resonator.

Several methods have been proposed for reducing motional resistance. According to Equation (5), the motional resistance of the capacitive MEMS resonator has a quadratic relationship with the gap, and reducing the gap between the electrode and the resonator can reduce the motional resistance [2]. To reduce motion resistance, Ref. [69] designed and fabricated an electrostatically driven DETF resonator with an electrode gap width of 1.03 μm. The test results show that the quality factor of the proposed resonator is approximately 30,000 in a vacuum environment, the resonance peak in an open loop is 18 dB, and the DC bias voltage is 20 V. However, this approach does have drawbacks in terms of manufacturing complexity. F. Ayazi and S. Pourkamali have fabricated thick device-layer MEMS resonators with a large transducing area and the nano-gap by the HARPSS fabrication process [70] and achieved a motion resistance below 1000 Ω [71]. Increasing the DC bias voltage can reduce the motional resistance in a linear relationship but worsens the pull-in instability and linearity. Alternatively, designing resonator arrays on the same substrate, and coupling the output current capacitively together can reduce the motional resistance. To form a coupled square resonator array for reduced series motional resistance and reduce the motional resistance by 5.9 times compared to the single resonator, Ref. [72] connected the corner of the plate of same frequency seven square resonators. However, the disadvantage is that the array performance is worsened by manufacturing tolerances and consumes a large amount of chip area. There are a number of other methods used to reduce resistance to motion, including incorporating high dielectric materials into the gap [73], two-dimensional coupling structures [74], cyclic coupling structures [75], and gap closure structures [76].

### 3.3. Frequency Accuracy

The frequency of MEMS resonators is shifted due to unavoidable manufacturing tolerances in the microfabrication process. The frequency accuracy and initial uniformity determine whether the designed resonator can be commercialized in timing applications. To improve accuracy and initial uniformity, it is necessary to correct the initial frequency of the resonator. Generally, mechanical trimming and electronic tuning are used to increase the frequency accuracy [7]. Mechanical trimming includes pulsed laser deposition, metal deposition or diffusion, local oxidation, and laser trimming. Electrical tuning mainly includes bias voltage tuning and phase-locked loop (PLL) tuning.

#### 3.3.1. Mechanical Trimming

The effective mass and stiffness determine the resonant frequency, so the resonant frequency can be trimmed by adding, removing, or changing the material of the resonator. An electronically controlled frequency trimming technique for the local thermal oxidation of a single crystal silicon resonator was illustrated [77]. When the MEMS resonator is biased with a relatively large current in an oxygen-rich environment, a thin layer of silicon dioxide can thermally grow on the silicon surface of the resonant structure. The change in structural stiffness results in a change in resonant frequency. Using the cooling effect of the resonator during resonance, automatic trimming can be achieved. The result shows that a frequency trimming effect of approximately 3.7% can be obtained [77]. A similar method of forming silicon-metal bonds by heating the deposited metal on the MEMS resonator to diffuse can also achieve trimming in the range of approximately 4000 ppm [78]. A MEMS resonant cavity with a top aluminum layer was proposed [79]. The −0.3 ppm/min to −12.2 ppm/min frequency trimming capability can be obtained by evaporating aluminum layer by heating. Directing the femtosecond laser beam to the resonator through a transparent cap or lid on the resonator package with varying the power and position of the laser can achieve a frequency accuracy of 2.6 ppm [80]. The laser trimming method for wineglass mode resonators is shown in Figure 4a,b, result showed that frequency split was significantly reduced to less than 0.5 Hz [81]. It may be worth noting that these operations are difficult to apply in the mass production of sensors and can only be selectively applied.

The mechanical trimming method can permanently change the resonant frequency and requires no additional power supply after completion. However, device surface residues may degrade the device performance and the regulation accuracy is limited. 

#### 3.3.2. Electrical Tuning

The resonator frequency can also be tuned by electrostatic spring softening [2]. A band electrostatic tuning scheme is proposed which was shown that the resonant frequency can be tuned in the range of 38 ppm by setting the bias voltage. It is equal to correcting the 0.25 μm linewidth error of the resonator [83]. D. E. Serrano fabricated AlN resonators with fully independent tuning potential using the substrate layer as a DC voltage electrode, achieving a frequency tuning range of 3100 ppm [84]. As shown in Figure 4c, when the tuning voltage increases from 5 V to 30 V, the resonator with a natural resonant frequency of 3990 Hz can obtain a tuning range of 3390 Hz to 3320 Hz [82]. A 228.42 Hz cantilever resonator was proposed by [85]. By adding nonlinear springs, the resonator has softening and hardening effects in different displacement directions, respectively. Therefore, the function of frequency tuning can be achieved by changing the position where the electrostatic force is applied. According to the simulation results, the device can output frequency signals from 124.2 Hz to 349.9 Hz within the voltage tuning range of 60V. The oscillator output frequency can also be varied using a high-performance fractional-N phase-locked loop. However, the PLLs may increase the power, phase noise, and chip area, which limits the application of PLLs for initial frequency correction of MEMS resonators. Generally, PLL is a common method in temperature compensation applications of MEMS resonators. 

Electrical tuning methods are more flexible and precise than mechanical trimming, but require additional power. However, the spring softening effect of electrostatically tuned springs is considered too weak to satisfy a wide range of initial frequency corrections. In addition to the above, a method of mechanically coupling the array is also proposed to improve the fabrication repeatability of the resonant frequency. A mechanically coupled array was used to achieve frequency averaging, and the overall standard deviation of the frequency is reduced with respect to the square root of the number of resonators in the array [86]. The result shows that the frequency standard deviation of an array consisting of three resonators can be 165.7 ppm [86].

### 3.4. Temperature Stability

The long-term stability of MEMS resonators is mainly affected by temperature. The spring softening effect by the elastic temperature coefficient (TCE) of silicon can result in a silicon MEMS resonator with a temperature coefficient of frequency (TCF) of approximately −30 ppm/K [87]. Thus, an uncompensated silicon MEMS resonator will have a frequency drift of up to −3750 ppm from −40 °C to 85 °C [88]. In particular, uncompensated silicon MEMS resonators can cause problems such as clock drift, signal errors, and loss of The Global Positioning System lock. This is the most important issue to overcome with silicon MEMS resonators replacing product grade quartz resonators. Therefore, temperature compensation of silicon MEMS resonators is necessary. The main temperature compensation schemes for silicon MEMS resonators can be classified as passive and active compensation.

#### 3.4.1. Passive Compensation

Passive compensation methods mainly include composite structures [89] and heavily doped silicon [90]. The composite structure approach uses positive TEC materials such as silicon dioxide (185 ppm/K) to neutralize the negative TEC of silicon. Taking the transverse bending resonant beam as an example [91], its resonant frequency can be expressed as:(18)$$f_{i}=\frac{\lambda_{i}}{2\pi L^{2}}\sqrt{\frac{E_{eq}I_{eq}}{\rho A}};\;i=1,2,3\cdots ,$$

The TCF can be defined as:(19)$$TCf=\frac{1}{f}\frac{df}{dT}=\frac{1}{2}\frac{1}{E_{eq}I_{eq}}\frac{d(E_{eq}I_{eq})}{dT}$$
and
(20)$$E_{eq}I_{eq}=\sum_{k}E_{k}I_{k}$$

The TCF can be calculated as:(21)$$TCf=\frac{1}{2}\frac{\sum_{k}E_{k}I_{k}TCE_{k}}{\sum_{k}E_{k}I_{k}}$$
where $\lambda_{i}$ is the mode constant, L is the length of the beam, and ρ is the density of the beam, A is the cross sectional area, $E_{eq}$ is the effective Young’s modulus of the material, $I_{eq}$ is the effective moment of inertia, and k is the number of the different materials.

A DETF resonator with a SiO_2_ coating [92] has a second order flip temperature similar to that of a quartz crystal. The flip temperature point is controlled by varying the ratio of the resonator’s SiO_2_ to silicon content. This means that a resonator with temperature stability at room temperature can be obtained by designing the inversion point. An extended mode MEMS resonator based on an oxide refilling process was proposed and reported in [93]. The first order TCF of the extended mode MEMS resonator can be compensated more effectively by placing silicon dioxide islands in high strain regions, resulting in a TCF of 4 ppm/K. AlN lamb wave resonators with negative TCF can also be temperature compensated by a composite structure [94]. The addition of a layer of SiO_2_ underneath the AlN achieved 250 ppm from −55 °C to 125 °C. A MEMS resonant gas sensor with oxide trenches on the edge of the cantilever, which has little degradation to the quality factor, was proposed by [95]. Experimental results show that the proposed design reduces the frequency temperature coefficient to 1.7 ppm/°C and the quality factor can reach 4700.

The TCE reduction can be achieved by introducing free carriers into the silicon lattice through doping to affect the silicon elasticity coefficient. That is, the elastic constants of silicon have a doping dependence. The advantage of silicon doping is that MEMS resonators can be fabricated on the highly doped silicon wafer and without any modifications to existing manufacturing processes. It also avoids the potential degradation of quality factors caused by composite structure stacking. Ref. [96] proposed a temperature compensation scheme using boron dopants for the silicon body acoustic resonator cavities (SiBAR). A significant reduction in TCF was measured at very high doping levels, thus verifying frequency-temperature dependence is affected by doping concentration [97,98]. According to [99,100], the frequency-temperature dependence is not only related to the doping concentration but also related to modes and orientations [88]. The experimental result shows that correctly positioned stretched mode resonators on highly n-doped silicon substrates have a TCF zero point, and that the total frequency variation of highly phosphorus-doped resonators aligned to <100> crystal orientations can reach less than 245 ppm over the −40 °C to 85 °C range. A 10 MHz highly doped capacitive MEMS resonator was reported by [101], and the frequency turnover point was changed by adjusting the crystal orientation. Results show less than ±16 ppm frequency jitter from −40 °C to 85 °C when the resonator is positioned 22.5° in the <110> direction. the temperature coefficient frequency of n-type doped silicon resonators in vibrational extensional mode was predicted by [102]. It is found that there is a frequency turnover point at room temperature at doping concentration levels of 1 × 10^19^ cm^−3^. The results also show that the greater the stress, the greater the frequency change of the MEMS resonator. a 105 MHz concave silicon bulk acoustic resonator (CBAR) with a linear TCF of −6.3 ppm/°C by boron doping was fabricated by [103]. Some highly doped silicon DETFs were fabricated and the effect of different alignment orientations on temperature stability were tested by [104]. The results show that a highly doped silicon resonator with proper alignment can achieve a 200 ppm frequency change from −35 °C to 85 °C.

In order to find suitable design rules and methods for temperature stable doped silicon resonators, Jaakkola proposed a prediction model combining sensitivity analysis, silicon elastic constant modeling, and free carrier theory [105,106]. The results demonstrate that the extensional modes, lamé modes, flexural modes, and torsional modes can all obtain low TCF when the resonator has a specific direction and doping concentration. Further, Refs. [99,107,108] fabricated a series of resonators with different doping concentrations, orientations, and resonance types, respectively. The elastic constants of the doped silicon resonator with temperature dependence were extracted by testing. Then, the temperature characteristic prediction model of the doped silicon resonator was established. The result showed that the prediction deviation was approximately 20 ppm [99]. Since temperature compensation requires tight control of the doping level, a new fabrication method for partially doped silicon resonators based on the epitaxial polysilicon encapsulation process was proposed [109]. This method avoids the lattice mismatch [110] and strain problems. An n-type degenerate doped resonator aligned with the <100> crystal orientation of the Si substrate was also proposed and achieved a TCF of −7.4 ppm/°C [111]. A piezoelectric AlN resonator combining a composite structure of heavily doped and oxide layers has been reported in recent years [112]. The proposed resonator can achieve ±21.5 ppm frequency variation from −40 °C to 85 °C. Table 4 compares different passive compensation methods.

Passive compensation can increase temperature stability. However, few MEMS resonators have been reported to achieve sub- ppm level frequency stability. This means that resonators that rely on passive compensation alone may not meet the stringent requirements of the most advanced wireless systems and precision navigation guidance applications.

#### 3.4.2. Active Compensation

Active compensation generally includes electrostatic forces, phase-locked loops, and oven control. The principle of electrostatic temperature compensation is similar to the electrical tuning method mentioned in the previous section, taking the parallel plate resonator as an example [113], the electrostatic tuning capability versus DC voltage applied to the MEMS resonator can be expressed as:(22)$$\frac{df}{f_{0}}=\frac{\varepsilon A_{e}}{k_{n}d^{3}}dV_{p}{}^{2}$$
where *A_e_* is the effective capacitive area, *k_n_* is the dynamic stiffness, and *d* is the initial capacitive gap.

A resonator with a shape similar to the letter “I” was proposed by Ref. [114]. By combining a diode array with a negative temperature coefficient and a charge pump, the frequency drift is only 240 ppm from 30 °C to 120 °C. A two-chip reference oscillator was demonstrated by [115]. A voltage multiplier circuit was used to generate a voltage of approximately 24 V, which was then applied to bias the resonator for temperature variations. The test result shows a temperature coefficient of 4.2 ppm/°C from 25 °C to 125 °C. As an improvement, parabolic correction schemes for 6 MHz, 10 MHz, and 20 MHz resonators were proposed with the 39 ppm temperature drift from 25 °C to 125 °C. Serrano fabricated the 32.768 kHz resonator on an AlN-on-SOI (Silicon-On-Insulator) substrate [116]. The effective capacitance area was increased by using an external frame combined with a rigid plate structure to achieve a frequency tuning range of 6400 ppm with only 6 V required. Test results have shown that the frequency drift of the resonator based on electrostatically tuned achieved 5 ppm from −25 °C to 100 °C. Composite structures combined with electrostatically tuned methods have also been proposed [117]. A DETF with a thermal oxide coating can achieve a small frequency variation (<110 ppm). In addition, the addition of electrostatic tuning on this basis can achieve ±2.5 ppm stability from −10 °C to 80 °C [117]. A 2.92 MHz free-free beam resonator on a 0.35um standard CMOS platform was integrated by [118], and an additional overhang electrode to provide electrical stiffness to simplify fabrication difficulty was introduced. The results show that the frequency drift of the proposed resonator decreases from 2120 ppm to 95 ppm in the range of 25 °C to 55 °C.

SiTime launched a unified packaged MEMS-CMOS programmable oscillator in 2007 [119]. The CMOS circuit includes a MEMS driver, temperature compensation, and a programmable frequency multiplier. It measures the temperature by a high-performance CMOS sensor to adjust the divider ratio for temperature compensation and achieves ±50 ppm frequency stability from −45 °C to 85 °C. Further, programmable MEMS thermistor-based temperature-to-digital converter (TDC) oscillators have been proposed [120]. Compared to conventional Bipolar Junction Transistor (BJT) temperature sensors, the thermistor is fabricated close to the resonator to achieve accurate thermal tracking, with a frequency stability of ±5 ppm from −40 °C to 85 °C. A fully integrated dual resonator was proposed in 2016 [121]. Temperature measurement is achieved in this architecture by measuring the frequency of two resonators. The sensors and resonators correspond, bringing a tight thermal coupling. Therefore, the resonator can maintain high frequency stability with rapid temperature fluctuations. The proposed integrated dual MEMS resonator achieved ±0.1 ppm stability from −45 °C to +105 °C. A piezoelectric resonator combining the composite structure and PLL compensation was reported in [122]. The composite structure oscillator with PLL achieves ±3 ppm frequency stability from −20 °C to +70 °C. Recently, Ref. [123] proposed to use of a fractional divider to compensate for the frequency drift of MEMS resonators. In addition, a short-term jitter suppression method based on digital-to-time converter modulation is proposed. The results show that the proposed method is more efficient than the PLL-implemented low-pass filter, and achieves a frequency drift of ±8 ppm.

Although temperature compensation methods such as composite structure, heavy doping, electrostatic and PLL have been proposed, these methods can only achieve ppm-level frequency stability [124], which is still difficult to satisfy high-end industries such as modern telecommunication systems [125] and military [126]. The oven controlled MEMS oscillator (OCMO) achieve ppb-level frequency stability by heating the resonator to a constant high temperature [124]. Silicon MEMS resonators can be integrated with heaters and sensors on a single chip to reduce power consumption, size, and cost [127], which provides a miniaturized solution for oven-controlled oscillators [126]. As shown in Figure 5a,b, Ref. [128] proposed a N-doping compensated LE mode resonator based on look-up table control method. First adjust the turnover point by changing the doping level, then fix the operating temperature at the turnover point through a control loop. The result shows that the proposed resonator achieves a stability of 0.5 ppm in the range of −35 °C to 85 °C. A 77.7 MHz lamé mode resonator using a structured resistor as the embedded temperature sensor and a silicon resistor as the heater was proposed by [127]. The oscillator relies on an automatic closed-loop control to regulate the temperature of a micro-oven and achieve a frequency stability of ±0.3 ppm across the temperature range of −25 °C to +85 °C. As an improvement, Ref. [124] proposed a 1.2 MHz oscillator consisting of a PLL and two DETF resonators with different temperature coefficients of frequency. Firstly, the temperature sensing is realized by tracking the difference frequency between two resonators, and then the active temperature compensation is realized by PLL controller. This method avoids the disadvantage that the accuracy of traditional temperature sensor may be affected by measuring circuit. Further, Ref. [129] proposed a novel dual-mode lamé resonator and a dual-mode DETF resonator. The proposed resonators achieved 1-week frequency stability close to 1.5 ppb by doping and active control. A dual-mode piezoelectric OCMO was proposed by [130], and it achieved a frequency stability of less than ± 400 ppb in the range of − 40 °C to 80 °C. Similarly, the ovenized dual-mode resonators based on highly doped single-crystal silicon were proposed in 2016 and 2019, and achieving ± 250 ppb [131] and ± 1.5 ppb [132] frequency offset over the 100 °C temperature range, respectively. A dual frequency resonator with an output frequency of 1.27 MHz or 13 MHz fabricated at 0 DEG and 45 DEG, respectively, was demonstrated by Ref. [133]. Temperature-controlled compensation is then achieved by tracking and locking the difference between the dual-mode frequencies, and the results show that the proposed resonator achieves stability of ±25 ppb in the range of −40 °C to 40 °C. The dual-mode single resonator can avoid the disadvantage of temperature detection error due to temperature gradient based on the dual resonator method. Table 5 summarizes the specific performance of some typical OCMOs. The OCMO has proven to be the most promising candidate for highly accurate oscillators [96]. In the future, OCMOs with sub-mW power consumption are also expected to be deployed in more applications such as sensor nodes, microsatellites or unmanned nanovehicles. However, today, OCMOs are still in the development stage from laboratory to commercial products. 

## 4. Summary and Future Perspective

In this paper, the working principles, important parameters, and key technologies of MEMS resonators are summarized by sorting out the proposed representative results. It aims to provide some design references for improving the performance of MEMS resonators. Although MEMS resonators have developed rapidly in recent years, there are still many problems to be solved. (1) There is a big difference in the choice of transduction mechanism. Capacitive resonators have problems, such as low electromechanical efficiency, narrow transduction gap, large DC bias, and precise process control requirements, which still need to be overcome. Piezoelectric MEMS resonators have less motional resistance and better power handling. Up to now, new piezoelectric materials are still being developed, and the application fields of piezoelectric MEMS resonators are still expanding. However, process standardization of piezoelectric resonators remains a challenge; (2) Temperature stability remains the most important issue facing MEMS resonators. Composite structures require changes to the existing process flow and may reduce the quality factor. The influence mechanism of heavy doping on temperature stability is complex and its effect is difficult to accurately predict. Electrostatic tuning is believed to have a limited effect on stiffer resonators. PLLs and OCMOs are limited by power and size, and thus cannot be widely used; (3) The common use of thin AlN films or overtone mode resonators at high frequencies can cause insufficient performance, such as high thermal resistance, large static capacitance, and low electromechanical coupling; and (4) As the structure of MEMS resonators becomes increasingly complex, problems such as nonlinearity [134] and chaotic phenomena [135] need to be gradually highlighted.

With the improvement of performance requirements for MEMS resonators and the broadening of application fields, the following are some studies that may become a focus in the future. (1) Numerical methods that capture specific energy loss mechanisms are becoming popular, such as the acoustoelectric loss due to the interaction of charge carriers with phonons [136]; (2) Combining a floating platform with the folded beam is a potential solution for OCMO structure. Additionally, crosstalk and mechanical coupling issues in dual-mode resonators need to be addressed; (3) Development of new materials and new models that meet 6G requirements, such as unique tangential LiNbO_3_ [137] and multimode resonators [138]; (4) Development of new MEMS devices that are highly immune to environmental drift, such as amplitude ratio output [139,140] and multi-mode excitation resonant sensor; (5) Flexible MEMS resonators [141] that can be used in the human body; (6) Monolithic integrated units for multiple sensors or multiple frequency outputs.

## Figures and Tables

**Figure 1 micromachines-13-02195-f001:**
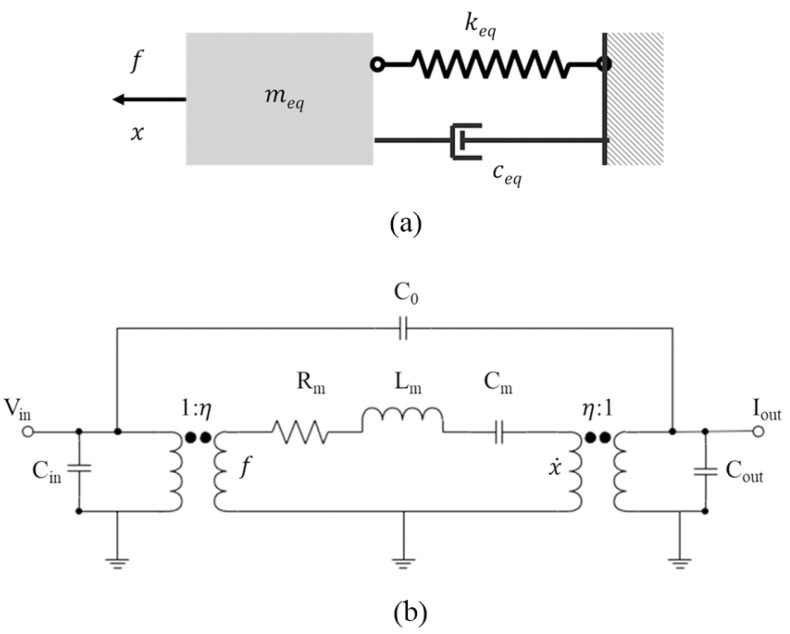
(**a**) Mass damping spring system; (**b**) Equivalent electrical model including feedthrough capacitor and parasitic capacitance.

**Figure 2 micromachines-13-02195-f002:**
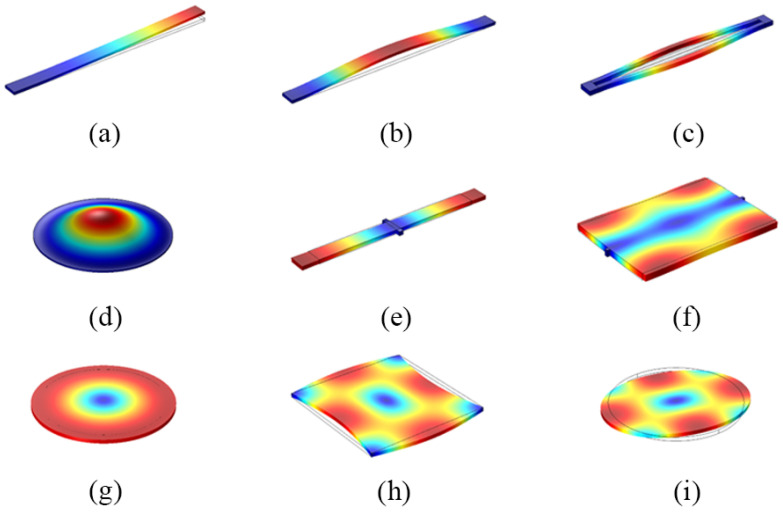
Diagrams of flexural modes (**a**–**d**) and bulk modes (**e**–**i**): (**a**) Single-ended fixed out-of-plane mode; (**b**) Double-ended fixed out-of-plane mode; (**c**) Double ended tuning forks (DETF) in-plane mode; (**d**) Above-membrane out-of-plane mode; (**e**) Length extension (LE) mode; (**f**) Width expansion (WE) mode; (**g**) Radial breathing mode; (**h**) Lamé mode; (**i**) Wine-glass mode, or face-shear (FS) mode when the resonator is rectangular.

**Figure 3 micromachines-13-02195-f003:**
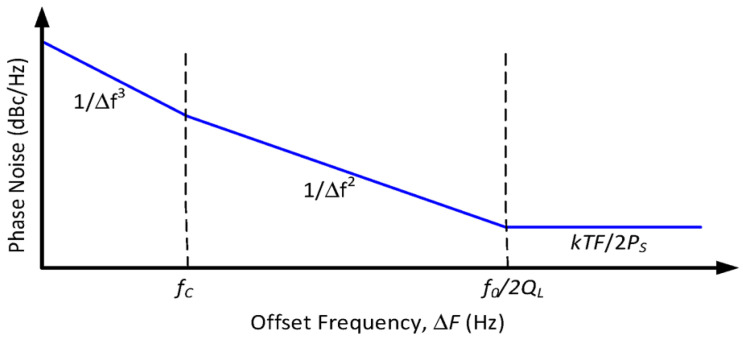
Typical phase noise power spectral density (single sided).

**Figure 4 micromachines-13-02195-f004:**
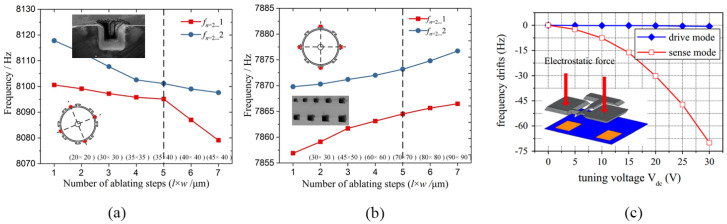
(**a**,**b**) When the laser ablation grooves are properly designed, the resonant frequency can be raised or lowered as desired [81]; (**c**) The change of the resonant frequency caused by the electrostatic force [82].

**Figure 5 micromachines-13-02195-f005:**
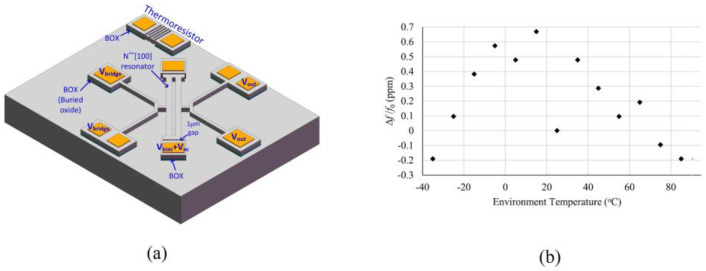
(**a**) Schematic view of the oven controlled resonator [128]; (**b**) The minimum drift of the resonant frequencies at the turnover point is measured to be less than 1  ppm in the temperature range of −35 °C to 85 °C [128].

**Table 1 micromachines-13-02195-t001:** Summary of frequency calculation formulas for common vibration modes.

Structure and Mode	Resonant Frequency	Parameter
Cantilever beam	$f_{0}=C_{n}\sqrt{\frac{E_{eq}}{\rho_{eq}}}\frac{t}{L^{2}}$	$f_{0}$ resonant frequency $C_{n}$ mode coefficient $E_{eq}$ modulus of elasticity $\rho_{eq}$ density t device thickness L device length n mode number W device width R device radius σ Poisson’s ratio D feature size D=L for LE mode D=W for WE mode and D=R for radial extension $G_{eq}$ shear modulus $C_{0}$ constant parameter $\kappa_{n}$ frequency parameter.
Double-clamped tuning fork	$f_{0}=\frac{{\left[(1+2n)\pi /2\right]}^{2}}{2\pi \sqrt{12}}\sqrt{\frac{E_{eq}}{\rho_{eq}}}\frac{W}{L^{2}}$	
Circular membrane	$f_{0}=\frac{10.22}{2\pi R^{2}\sqrt{12\rho_{eq}\left(1-\sigma^{2}\right)/E_{eq}t^{2}}}$	
Extension mode	$f_{0}=\frac{n}{2D}\sqrt{\frac{E_{eq}}{\rho_{eq}}}$	
FBAR	$f_{0}=\frac{1}{2t}\sqrt{\frac{E_{eq}}{\rho_{eq}}}$	
Lamé	$f_{0}=\frac{1}{\sqrt{2}L}\sqrt{\frac{G_{eq}}{\rho_{eq}}}$	
Face shear	$f_{0}=\frac{C_{0}}{L}\sqrt{\frac{G_{eq}}{\rho_{eq}}}$	
Wineglass	$f_{0}=\frac{\kappa_{n}}{2\pi R}\sqrt{\frac{E_{eq}}{\rho_{eq}\left(1-\sigma^{2}\right)}}$	

The formula is referenced from literature [7].

**Table 2 micromachines-13-02195-t002:** Summary of representative different modes resonators in specific applications.

Type	Frequency	Q	Pressure (mTorr)	Transmission (dB)	Vibration Modes	Reference	Schematic Illustration
Capacitive	32.768 kHz	~15,000	50	−74	Flexural	[22]	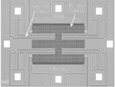
Capacitive	6.35 MHz	1,700,000	0.15	−17	Lamé	[26]	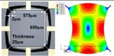
Capacitive	51.3 MHz	128,400	0.08	−90	Lamé	[27]	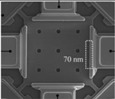
Capacitive	107.3 MHz	11,000	Standard atmosphere	−80	Whispering gallery	[28]	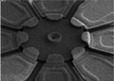
Capacitive	150.9 MHz	18,000	0.225	−72	Radial-contour	[24]	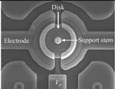
Piezoelectric	10 MHz	4682	Standard atmosphere	−20	Width expansion	[29]	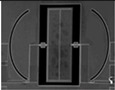
Piezoelectric	14.02 MHz	5000	~mTorr	−24	Length extension	[23]	
Piezoelectric	48.14 MHz	10,000	~mTorr	−8	Width expansion	[23]	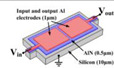
Piezoelectric	52 MHz	4743	Standard atmosphere	−25	Lateral-extension	[30]	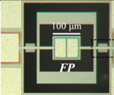
Piezoelectric	882 MHz	220	Standard atmosphere	−46	Contour mode	[31]	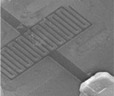

**Table 3 micromachines-13-02195-t003:** Typical methods of improving quality factor.

Frequency	Mode	Type	Original Q	Enhanced Q	Methods	Reference	Schematic Illustration
52 MHz	Lateral-extension	Anchor loss	606	4743	Frame structure with PnC	[30]	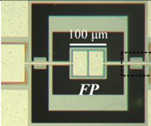
51.3 MHz	Lamé	Anchor loss	56,400	128,400	The beam with root slots	[27]	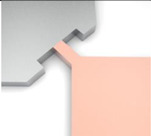
10.03 MHz	Lateral mode	Anchor loss	2618	3945	Reflective structures	[41]	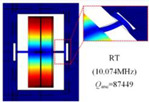
10.03 MHz	Lateral mode	Anchor loss	2618	4522	PnC	[41]	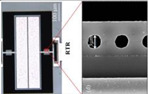
10 MHz	Width expansion	Anchor loss	1570	4682	PnC + Reflector	[29]	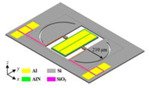
610 kHz	Flexural mode	TED	13,000	16,000	Slots	[50]	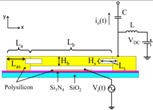
400 kHz	Flexural mode	TED	15,000	40,000	Slots	[51]	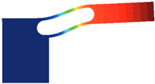
20 kHz	Flexural mode	Coating loss	3000	8000	Coating coverage	[52]	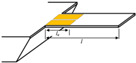

**Table 4 micromachines-13-02195-t004:** Typical methods of passive compensation.

Frequency (MHz)	Type	Methods	Reference	Stability
0.39	In-plane flexural	SiO_2_	[95]	1.7 ppm/°C [10 °C to 90 °C]
1	DETF	SiO_2_	[89]	−0.02 ppm/°C^2^ [−55 °C to 125 °C]
1.024	DETF	SiO_2_	[92]	−0.02 ppm/°C^2^ [−40 °C to 120 °C]
711	Lamb Wave	SiO_2_	[94]	−0.021 [−55 °C to 125 °C]
0.47	DETF	Doping	[104]	190 ppm [5 °C to 85 °C]
9	Lateral extensional	Doping	[101]	±20 ppm [−40 °C to 85 °C]
10	square extensional	Doping	[101]	±16 ppm [−40 °C to 85 °C]
23	Extensional mode	Doping	[90]	10 ppm [−40 °C to 85 °C]
25.09	Lateral extensional	Doping	[88]	245 ppm [−40 °C to 85 °C]
24.44	Width extensional	Doping and SiO_2_	[112]	±21.5 ppm [−40 °C to 85 °C]

**Table 5 micromachines-13-02195-t005:** Typical methods of active compensation.

Frequency (MHz)	Type	Methods	Reference	Stability
2.92	Free-free beam	Electrostatic	[118]	0.44 ppm/°C 25 °C to 55
5.5	I-shaped bulk	Electrostatic	[11]	39 ppm 25 °C to 125 °C
1.126	DETF	SiO_2_+ electrostatic	[117]	±2.5 ppm −10 °C to 80 °C
0.54	In-plane flexural	Doping and Single-Temperature Calibration	[123]	±8 ppm 5 °C to 85 °C
77.7	Lamé mode	Oven control	[127]	±0.3 ppm −25 °C to 85 °C
1.2	DETF	Oven control	[124]	±1 ppm −20 °C to 80 °C
1.2	DETF	Calibration and control	[124]	±0.05 ppm −20 °C to 80 °C
1.2	Plate Bending	Doping and control	[133]	±25 ppb −40 °C to 40 °C
10	Length-extensional	Doping and control	[128]	±0.5 ppm −35 °C to 85 °C
13	Lamé	Doping and control	[133]	±5 ppb −40 °C to 40 °C
42.7	Shear mode	Doping and control	[130]	±0.4 ppm −40 °C to 80 °C

## Data Availability

Not applicable.

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
