# Peer review of "Concepts and Key Technologies of Microelectromechanical Systems Resonators"

_micromachines, 2022, doi:10.3390/mi13122195_

Round 1

Reviewer 1 Report

1. Lines 93-95: Are formulas 4-6 the result of the authors' research? If not, please provide a link to the source.

2. Line 137: Figure 3a shows a capacitive resonator. However, all structural elements are fixed, and therefore there are no moving parts. Is it supposed to be like this or is it an image error?

3. Lines 351-353: The authors write "The -0.3 ppm/min to -12.2 ppm/min frequency trimming capability can be obtained by evaporating aluminum layer by heating. Directing the femtosecond laser beam to the resonator through a transparent cap or lid on the resonator package...". It may be worth noting that these operations are difficult to apply in the mass production of sensors and can only be applied selectively.

4. Lines 369-370: The authors write "As shown in Figure 5c, when the tuning voltage increases from 5 V to 30 V, the resonant frequency can change by 70 Hz". Perhaps it would be better to indicate not the absolute value of the change, but to express it as a percentage of the sonic frequency. Or give the value of the main frequency for the convenience of readers.

Author Response

Cover letter

Dear Reviewer,

Thanks very much for taking your time to review this manuscript.

Manuscript ID: micromachines-2055706

Type of manuscript: Review

On behalf of my co-authors, we thank you for giving us a chance to revise and improve the quality of our article. We have read the comments carefully and have made revisions which are marked in red in the paper. We have tried our best to revise our manuscript according to the comments and these changes will not influence the content and framework of the paper. We really appreciate all your generous Points and suggestions! Please find my itemized responses below and my revisions in the re-submitted files.

Response to Reviewer 1 Comments

Point 1: Lines 93-95: Are formulas 4-6 the result of the authors' research? If not, please provide a link to the source.

Response: We are very sorry for overlooking the link to the source of formulas 4-6, and it is added at Line 103. In addition, we consolidated the frequency-related formulas into Table 1 to enhance the readability of the manuscript and hope that it is now clearer. (line 103)

Point 2:  Line 137: Figure 3a shows a capacitive resonator. However, all structural elements are fixed, and therefore there are no moving parts. Is it supposed to be like this or is it an image error?

Response: It is not an image error. We apologize for not expressing the capacitive resonator clearly. The plate capacitive resonator shown in Figure 3a is a double-terminal fixed structure, and the movable part refers to the rigid plate center that moves up and down during resonance similar to Figure 3b. Also, considering that the basic principles have been widely discussed in many different papers, we have removed Figure 3 to prevent additional confusion for readers (lines 155-158).

Point 3:  Lines 351-353: The authors write "The -0.3 ppm/min to -12.2 ppm/min frequency trimming capability can be obtained by evaporating aluminum layer by heating. Directing the femtosecond laser beam to the resonator through a transparent cap or lid on the resonator package...". It may be worth noting that these operations are difficult to apply in the mass production of sensors and can only be applied selectively.

Response: Your Points are very important to improve the integrity of this manuscript. With reference to your suggestion, we added a Point on line 442: "It may be worth noting that these operations are difficult to apply to mass production of sensors and can only be applied selectively." to provide design reference for readers. (line 442)

Point 4:  Lines 369-370: The authors write "As shown in Figure 5c, when the tuning voltage increases from 5 V to 30 V, the resonant frequency can change by 70 Hz". Perhaps it would be better to indicate not the absolute value of the change but to express it as a percentage of the sonic frequency. Or give the value of the main frequency for the convenience of readers.

Response: Thank you for your suggestion. With reference to your Points, we have revised the original text “the resonant frequency can change by 70” of line 460 to " the resonator with a natural resonant frequency of 3990Hz can obtain a tuning range of 3390Hz to 3320Hz”. It is explained that the change range of the main frequency replaces the absolute value of the change to understand its adjustment ability more intuitively. (line 460)

All revision traces can be viewed in the revision mode of the word file. We would like to thank the referee again for taking the time to review our manuscript and hope that the correction will meet with approval. Should you have any questions, please contact us without hesitation.  And special thanks to the editors and reviewers for your good comments again. I wish this revision will be acceptable for publication in your journal.

Yours Sincerely,

Tianren Feng1

Quan Yuan1

Duli Yu1

Bo Wu2

Hui Wang2

December 1, 2022

1 College of Information Science and Technology, Beijing University of Chemical Technology, Beijing 100029, China

2 Guangdong Institute of Semiconductor Micro-nano Manufacturing Technology, Guangdong 528225, China

E-mail: [email protected]; [email protected]

Reviewer 2 Report

This paper did a review on the MEMS resonators. The basic principles and techniques for performance optimization are provided. The paper tried to promote the development of MEMS resonators, also provide a scientific and technical support for the improvement of MEMS resonators to practical system. However, I do not think this paper well fulfill this task. My main comments are as following.

1. About the contribution of this paper. Resonator is a classical device in MEMS, and many relevant reviews has been published. A simple comparison between the submitted paper and recent published ones should be done to clearly show the contribution and value of this paper. Moreover, the basic principles and performance parameters have been widely discussed in many different papers, and many conclusions have been written in textbook. It is not necessary to give long paragraphs for them.

2.As the conclusion says” MEMS resonators have experienced rapid development in recent years”. However, most of the cited paper are published about 15 years ago, which mat have been referred by other reviews. The recent progresses are totally missing.

3. The application of MEMS resonator is an important aspect to determine the its design and optimization. However, the application is not mentioned in this paper, which cannot well reflect the target of each optimization technique.      

4. In this paper, the existing problems and prediction of future trend are not discussed. It is strongly suggested to strengthen this part.

5. Thermal and magnetic excitations are not included in the principles section. They may be not as strong as the mentioned mechanisms, but also received many attentions.

Author Response

Cover letter

Dear Reviewer,

Thanks very much for taking your time to review this manuscript.

Manuscript ID: micromachines-2055706

Type of manuscript: Review

On behalf of my co-authors, we thank you for giving us a chance to revise and improve the quality of our article. We have read the comments carefully and have made revisions which are marked in red in the paper. We have tried our best to revise our manuscript according to the comments and these changes will not influence the content and framework of the paper. We really appreciate all your generous Points and suggestions! Please find my itemized responses below and my revisions in the re-submitted files.

Response to Reviewer 2 Comments

Point 1: About the contribution of this paper. Resonator is a classical device in MEMS, and many relevant reviews has been published. A simple comparison between the submitted paper and recent published ones should be done to clearly show the contribution and value of this paper. Moreover, the basic principles and performance parameters have been widely discussed in many different papers, and many conclusions have been written in textbook. It is not necessary to give long paragraphs for them.

Response: It is really a good idea as the Reviewer suggested, and we have changed them to meet the Reviewer’s thoughts.

1.Considering the Reviewer’s suggestion, we have added a brief comparison with recently published review papers in lines 43-57.

For example, [1] discusses the vibration modes of MEMS resonators, simplified models, and their applications. [2] provides an overview of the fabrication methods for silicon-based MEMS resonant sensors. [3] presents a review of recent advances in resonator-based M/NEMS logic devices. [4] discusses the piezoelectric resonator materials, process flow, and performance improvement methods. [5] mainly reviews the dissipation analysis methods and quality factor enhancement strategies of piezoelectric MEMS lateral vibration resonators. In contrast to recent reviews, our manuscript summarizes the results of the design-to-engineering flow of MEMS resonators for timing, frequency, and resonance sensing applications. These include quality factor improvement methods and temperature compensation methods in design, as well as mechanical trimming, electrical tuning, and temperature control in engineering. These contents are not found in other mentioned articles. And it aims to provide some design references for improving the performance of MEMS resonators.

2.We agree with the Point: "Many conclusions have been written in the textbook. It is not necessary to give long paragraphs for them." And we think that some brief introductions and conclusions are helpful for the organization of the article. Therefore, we have condensed the manuscript. For example, the frequency-related formulas (lines 110-118, lines 126-128, lines 134-135, lines 140-144) into Table 1 and removed some relevant content to streamline (lines 83-84, lines 155-158) the manuscript and hope it is now clearer.

Please see the revised manuscript, lines 60–70, lines 97–98, line 103, lines 110–118, lines 126–128, lines 134–135, lines 140–144, lines 156–159, lines 171–172, lines 156–159, and lines 213–218 for details.

[1].Platz, D.; Schmid, U. Vibrational Modes in MEMS Resonators. J. Micromech. Microeng. 2019, 29, 123001, doi:10.1088/1361-6439/ab4bad.

[2].Verma, G.; Mondal, K.; Gupta, A. Si-Based MEMS Resonant Sensor: A Review from Microfabrication Perspective. Microelectronics Journal 2021, 118, 105210, doi:10.1016/j.mejo.2021.105210.

[3].Ilyas, S.; Younis, M.I. Resonator-Based M/NEMS Logic Devices: Review of Recent Advances. Sensors and Actuators A: Physical 2020, 302, 111821, doi:10.1016/j.sna.2019.111821.

[4].Pillai, G.; Li, S.-S. Piezoelectric MEMS Resonators: A Review. IEEE Sensors Journal 2021, 21, 12589–12605, doi:10.1109/JSEN.2020.3039052.

[5].Tu, C.; Lee, J.E.-Y.; Zhang, X.-S. Dissipation Analysis Methods and Q-Enhancement Strategies in Piezoelectric MEMS Laterally Vibrating Resonators: A Review. Sensors 2020, 20, 4978, doi:10.3390/s20174978.

Point 2: As the conclusion says” MEMS resonators have experienced rapid development in recent years”. However, most of the cited paper are published about 15 years ago, which mat have been referred by other reviews. The recent progresses are totally missing.

Response: We appreciate it very much for this good suggestion, and we have done it according to your ideas. To make this paper more comprehensive, a review of some recent representative-related studies is added. Please see the revised manuscript, lines 43–49, lines 146 Table2, lines 179-193, lines 244-247, lines 280-282, lines 284-299, line 302, lines 327-341, lines 356-368, lines 396-400, lines 461-466, lines 514-517, lines 532-544, line 568, lines 593-597, lines 613-618, lines 646-651, and line 662. Now, the revised manuscript reviews a total of 100 papers in five years, including 64 papers in the last three years.

Point 3: The application of MEMS resonator is an important aspect to determine the its design and optimization. However, the application is not mentioned in this paper, which cannot well reflect the target of each optimization technique.

Response: We thank the reviewer for pointing this out. In the text, we indicate the field of application of the content described. Please see the revised manuscript, line 15, line 18, line 194, line 205, and line 117. Meanwhile, there are also some descriptions of specific applications in the original manuscript. They are in line 130, line 134, line 379, and line 486 of the original manuscript.

Point 4: In this paper, the existing problems and prediction of future trend are not discussed. It is strongly suggested to strengthen this part.

Response: We are extremely grateful to the Reviewer for this suggestion. According to your suggestion, we changed [conclusion] to [summary and future perspective] (line 664). Added discussion of existing problems and future development trends. Please see the revised manuscript, lines 664-701.

Point 5: Thermal and magnetic excitations are not included in the principles section. They may be not as strong as the mentioned mechanisms, but also received many attentions.

Response: Thanks for your suggestion, it is very helpful to increase the comprehensiveness of the manuscript. We have added a discussion of thermally and magnetically actuated MEMS resonators, please see lines 179-193 of the revised manuscript. However, the focus of this manuscript remains on the discussion of performance optimization, and thus thermally and magnetically actuated MEMS resonators are briefly introduced.

All revision traces can be viewed in the revision mode of the word file. We would like to thank the referee again for taking the time to review our manuscript and hope that the correction will meet with approval. Should you have any questions, please contact us without hesitation.  And special thanks to the editors and reviewers for your good comments again. I wish this revision will be acceptable for publication in your journal.

Yours Sincerely,

Tianren Feng1

Quan Yuan1

Duli Yu1

Bo Wu2

Hui Wang2

December 1, 2022

1 College of Information Science and Technology, Beijing University of Chemical Technology, Beijing 100029, China

2 Guangdong Institute of Semiconductor Micro-nano Manufacturing Technology, Guangdong 528225, China

E-mail: [email protected]; [email protected]

Round 2

Reviewer 2 Report

The revised work is good, and the paper can be accepted now.